# High-throughput analysis of lung immune cells in a combined murine model of agriculture dust-triggered airway inflammation with rheumatoid arthritis

Rohit Gaurav[1]*, Ted R. Mikuls[2,3], Geoffrey M. Thiele[2,3], Amy J. Nelson[1], Meng Niu[4], Chittibabu Guda[4], James D. Eudy[4], Austin E. Barry[1], Todd A. Wyatt[2,5,6,7], Debra J. Romberger[2,5], Michael J. Duryee[2,3], Bryant R. England[2,3], Jill A. Poole[1]

1 Division of Allergy and Immunology, Department of Internal Medicine, University of Nebraska Medical Center, Omaha, NE, United States of America, 2 Veterans Affairs Nebraska-Western Iowa Health Care System, Research Service, Omaha, NE, United States of America, 3 Division of Rheumatology & Immunology, Department of Internal Medicine, University of Nebraska Medical Center, Omaha, NE, United States of America, 4 Department of Genetics, Cell Biology and Anatomy, University of Nebraska Medical Center, Omaha, NE, United States of America, 5 Division of Pulmonary, Critical Care & Sleep, Department of Internal Medicine, University of Nebraska Medical Center, Omaha, NE, United States of America, 6 Department of Environmental, Agricultural & Occupational Health, College of Public Health, University of Nebraska Medical Center, Omaha, NE, United States of America, 7 Department of Internal Medicine, University of Nebraska Medical Center, Omaha, NE, United States of America

* rohit.gaurav@unmc.edu

**Data Availability Statement:** The raw and analyzed data is deposited in the Gene Expression Omnibus (GEO) database with access number GSE155436.

## Abstract

Rheumatoid arthritis (RA)-associated lung disease is a leading cause of mortality in RA, yet the mechanisms linking lung disease and RA remain unknown. Using an established murine model of RA-associated lung disease combining collagen-induced arthritis (CIA) with organic dust extract (ODE)-induced airway inflammation, differences among lung immune cell populations were analyzed by single cell RNA-sequencing. Additionally, four lung myeloid-derived immune cell populations including macrophages, monocytes/macrophages, monocytes, and neutrophils were isolated by fluorescence cell sorting and gene expression was determined by NanoString analysis. Unsupervised clustering revealed 14 discrete clusters among Sham, CIA, ODE, and CIA+ODE treatment groups: 3 neutrophils (inflammatory, resident/transitional, autoreactive/suppressor), 5 macrophages (airspace, differentiating/recruited, recruited, resident/interstitial, and proliferative airspace), 2 T-cells (differentiating and effector), and a single cluster each of inflammatory monocytes, dendritic cells, B-cells and natural killer cells. Inflammatory monocytes, autoreactive/suppressor neutrophils, and recruited/differentiating macrophages were predominant with arthritis induction (CIA and CIA+ODE). By specific lung cell isolation, several interferon-related and autoimmune genes were disproportionately expressed among CIA and CIA+ODE (e.g. *Oasl1*, *Oas2*, *Ifit3*, *Gbp2*, *Ifi44*, and *Zbp1*), corresponding to RA and RA-associated lung disease. Monocytic myeloid-derived suppressor cells were reduced, while complement genes (e.g. *C1s1* and *Cfb*) were uniquely increased in CIA+ODE mice across cell populations. Recruited and inflammatory macrophages/monocytes and neutrophils expressing interferon-, autoimmune-, and complement-related genes might contribute towards pro-fibrotic inflammatory

**Funding:** RG is supported by Central States Center for Agricultural Safety and Health NIOSH (U54OH010162). Study supported by grants from the National Institute of Environmental Health Sciences (R01ES019325 to JAP), National Institute for Occupational Safety and Health (U54OH010162 to JAP and TAW). TAW is the recipient of a Research Career Scientist Award (IK6 BX003781) from the Department of Veterans Affairs. TRM is supported by VA (CX000896 and BX004600) and grants from the National Institute of General Medical Sciences (U54GM115458) and the National Institute on Alcohol Abuse and Alcoholism (R25AA020818). BRE is supported by grants from the National Institute of General Medical Sciences (U54GM115458) and the Rheumatology Research Foundation. DJR is supported by VA Merit review 5I01CX001714-02. Study also supported by the Fred & Pamela Buffett Cancer Center Shared Resource, supported by the National Cancer Institute under award number P30CA036727. The University of Nebraska DNA Sequencing Core and Bioinformatics Core receive partial support from the National Institute for General Medical Science (NIGMS) NE-INBRE (Nebraska Research Network in Functional Genomics) (P20GM103427) and COBRE (1P30GM110768). This publication's contents are the sole responsibility of the authors. The funders had no role in study design, data collection and analysis, decision to publish, or preparation of the manuscript.

**Competing interests:** The authors have declared that no competing interests exist.

lung responses following airborne biohazard exposures in setting of autoimmune arthritis and could be predictive and/or targeted to reduce disease burden.

## Introduction

Several lung diseases have been associated with rheumatoid arthritis (RA), including interstitial lung disease (ILD), chronic obstructive pulmonary disease (COPD), pulmonary nodules, pleural effusions, bronchiolitis obliterans, and asthma [1–3]. Affecting up to 40% or more of RA patients, RA-associated lung diseases pose a substantial burden to healthcare systems because of the increased morbidity and mortality, decreased quality of life, and tremendous healthcare costs [2, 4, 5]. Evidence of RA-related autoantibodies generated in lung mucosa, even in the absence of articular manifestations of RA [6], as well as increased concentrations of serum anti-citrullinated protein antibody accompanying RA-related lung diseases [1, 4, 7], reinforces the pathogenic links between pulmonary inflammation and autoimmunity leading to the development of RA. Therapeutic options for RA-associated lung disease are limited [8], and key cellular and/or mediators predictive of the development and/or progression of RA-associated lung disease are lacking [9]. Thus, studies are warranted to investigate and identify precise mechanisms underpinning these associations.

Exposure to environmental factors such as cigarette smoke represent shared risk factors in the development of RA and inflammatory lung diseases [3, 10]. However, insight into how inhalant injury might lead to or exacerbate RA and its pulmonary manifestations, has been limited in the absence of a relevant disease model. Recently a pre-clinical animal model to provide insight into the important cellular players and decipher molecular and potential mechanistic pathways involved in RA-associated inflammatory lung disease was established [11]. Specifically, the combination of the collagen-induced arthritis (CIA) model with a model of airborne biohazard exposure (e.g. agriculture related-organic dust extract/ODE) resulted in augmented arthritis inflammatory score and bone deterioration, increased systemic autoimmunity with increased anti-cyclic citrullinated peptide IgG antibodies, and promotion of pre-fibrotic inflammatory lung changes in mice [11] consistent with RA-associated lung disease pathophysiology. However, the mechanisms underlying these observations are not known. Here, we hypothesized that RA-associated lung disease is associated with unique cellular phenotypes and specific novel gene expression of *in vivo* exposed lungs. Leveraging this novel murine model, single-cell RNA sequencing (scRNA-seq) and unsupervised clustering were applied to lung immune cells among Sham, CIA, ODE, and CIA+ODE treatment groups to explore exposure-related differences in cellular subsets, transcriptional profiles, and associated biologic pathways. In separate complimentary studies to confirm key scRNA-seq findings, lung myeloid-derived cells (i.e. monocytes/macrophages and granulocytes) were isolated and subjected to gene-expression analysis.

## Materials and methods

### Animals

Arthritis prone DBA/1J male mice between 6–8 weeks of age were purchased from Jackson Laboratory (Bar Harbor, ME) and fed alfalfa-free chow *ad libitum* (Envigo Teklad, Huntingdon, Cambridgeshire, UK) as per supplier recommendations. All animal procedures were approved by the UNMC Institutional Animal Care and Use Committee (protocol #19-043-05) and were in accordance with NIH guidelines for the use of rodents and has been described

previously [11]. All procedures on mice were done under isofluorane to minimize distress. After every instillation or injection, animals were monitored consistently until they regained consciousness and mobility. They were monitored every day once by the investigators and once by the vivarium staff for any signs of discomfort or distress.

## Organic dust extract

Organic dust extract (ODE) was prepared as previously reported [12] to model airway inflammatory disease. Briefly, an aqueous extract of organic dust from swine confinement feeding facilities (microbial-enriched agriculture setting) was prepared by incubating 1 g dust in 10 ml sterile Hank's Balanced Salt Solution (Mediatech, Manassas, VA) for 1 hour at room temperature followed by centrifugation for 10 minutes at 2,850 x g and repeated twice. The end supernate was filter-sterilized with a 0.22 µm syringe filter to remove any microorganisms and coarse particles. Constituents of the extract have been well characterized and include both endotoxin and peptidoglycans [11, 12]. ODE stock was prepared and stored at –20˚C in batches; aliquots were diluted for each experiment to a final concentration (vol/vol) of 12.5% in sterile phosphate buffered saline (PBS; pH = 7.4). Endotoxin concentrations ranged from 150–175 EU/mL as determined using the limulus amebocyte lysate assay (Lonza, Walkersville, MD). This concentration of ODE has been previously shown to produce optimal experimental effects and is well-tolerated in mice [11, 12].

## Animal co-exposure model

The protocol for the co-exposure model has been previously described [11]. Briefly, mice were age-matched and randomized to 4 treatment groups: Sham (saline injection, saline inhalation), collagen-induced arthritis (CIA; CIA injection, saline inhalation), ODE (saline injection, ODE inhalation), and CIA + ODE (CIA injection, ODE inhalation). CIA was induced with two subcutaneous tail injections (100 µg) of chick type II collagen (2 mg/ml) emulsified in Freund's complete adjuvant (FCA) on day 1 and in Freund's incomplete adjuvant (IFA) on day 21. Sham injections and saline inhalation were conducted with sterile PBS. Following an established protocol, 50 µl of intranasal saline or 12.5% ODE daily for 5 weeks (weekends excluded) was used to induce airway inflammatory disease [11, 12]. Mouse treatment groups were ran in parallel with euthanization occurring 5 weeks after initiation of treatments.

## Arthritis evaluation

Arthritis inflammatory scores were assessed weekly using the semiquantitative, mouse arthritis scoring system provided by Chondrex (www.chondrex.com) as previously described [11]. Scores range from 0 (no inflammation) to 4 (erythema and severe swelling encompassing ankle, foot, and digits). Arthritis evaluation was assessed on 8 mice per treatment group from 3 independent studies.

## Lung histopathology

Lung sections of Sham, CIA, ODE, and CIA+ODE treatment groups previously obtained [11] were stained with H&E or with anti-CD3 (1:100, Cat#ab5690, Lot#GR3356033-2), anti-CD68 (1:50, Cat#ab31630, Lot#GR3305929-3), and anti-MPO (1:25, Cat#ab9535, Lot#GR331736-4) from Abcam (Cambridge, MA), anti-CD45R (1:40, Cat#14-0452-82, Lot#2178350) from Invitrogen (Grand Island, NY), and anti-CCR2 (1:100, Cat# NBP267700, Lot# HMO537) from Novus Biologicals (Centennial, CO). Cross absorbed (H+L) goat anti-rabbit (Cat#A32731, Lot#UK290266), goat anti-mouse (Cat#A32727, Lot#UL287768) and goat-anti rat

(Cat#A21434, Lot#2184321) from Thermo Fisher, Grand Island, NY) were used at 1:100 dilution as secondary antibodies. Slides were mounted with VECTASHIELD® Antifade Mounting Medium with DAPI (Cat#H-1200, Lot#ZG1014, Burlingame, CA) and visualized under Zeiss fluorescent microscope.

## Single-cell RNA sequencing

For these studies, 2 mice per treatment group were euthanized with isoflurane in a desiccator. Lungs were exposed from the thoracic cavity and perfused with 10 ml heparin-PBS [11]. Harvested lungs were dissociated with gentleMACS dissociator (Miltenyi Biotech, Auburn, CA) in a digestion solution (collagenase I, 0.2 µg/µl + DNase I, 75 U/ml + heparin, 1.5 U/ml, in Dulbecco's Modified Eagle's Media; DMEM) and incubated for 30 minutes at 37°C in a shaking incubator. Digestion solution activity was neutralized with PBS containing 4 mM EDTA. Red blood cells were lysed with 1 ml ammonium-chloride-potassium (ACK) lysis buffer (Quality Biological, Gaithersburg, MD) for 1 minute and neutralized with ice-cold DMEM (Gibco). Cells were processed for RNAseq in FACS buffer (2% fetal bovine serum (FBS) + 0.1% NaN$_3$ in PBS). All reagents purchased from Sigma unless otherwise specified.

Single cell suspensions generated from whole lung were quantified and viability tested using a LUNA-FL$^{TM}$ Dual Fluorescence Cell Counter (Logos Biosystems, Annandale, VA). Single cells were then isolated from cell suspensions (100–2,000 cells/µl) using a 10x Chromium controller per manufacturer's suggested protocol (10x Genomics, Pleasanton, CA). Following cell capture, the gel beads in emulsion (GEM)/sample solution was recovered and placed into strip tubes. Reverse transcription was performed on a thermocycler (C1000 Touch™ Thermal Cycler, Bio-Rad, Hercules, CA) per recommended protocol followed by cDNA amplification. Amplified products were solid phase reversible immobilization (SPRI) bead-purified and evaluated by Fragment Analyzer (Agilent, Santa Clara, CA). Twenty-five percent of the cDNA volume was subjected to fragmentation and double-sided SPRIselect (Beckman Coulter, Indianapolis, IN) was used for PCR purification and clean-up. After adaptor ligation, SPRI clean-up was performed and PCR amplification using sample specific indexes for each sample was completed. PCR products were purified, quantified and library size distribution determined by Fragment Analyzer. Libraries were sequenced per the manufacturer's suggested parameters on a NextSeq500 sequencer to an average depth of 50,000 reads per cell.

## Single-cell RNA sequencing data processing

Basecall files (BCL) were generated through 10xGenomics Chromium Single cell 3' Solution followed by RNA Sequencing using Nextseq 500 and Nextseq 550. *Cellranger mkfastq* was used for demultiplexing and to convert BCL files into FASTQ files. FASTQ files were run through *Cellranger count* to perform alignment (using STAR aligner), filtering, and unique molecular identifier (UMI) counting. Chromium cellular barcodes were used to generate gene-barcode matrices, perform clustering, and do gene expression analyses. *Cellranger aggr* was used to normalize and pool the results from different samples, followed by the application of Principal Components Analysis (PCA) to change the dimensionality of the datasets. t-SNE (t-Stochastic Neighbor Embedding) was used to visualize the data in a 2-D space. Graph-based unsupervised clustering was then used to cluster the cells. We used Loupe browser [13], R packages including cellranger R-kit [14], complex heatmap [15], and Geom_violin [16] for more in-depth analysis to compare genes expression in each cluster compared to all the other clusters and plot the data. The data sets have been deposited to the Gene Expression Omnibus (GEO) database with access number GSE155436.

## Lung cell sorting

In separate studies with 8 mice per treatment group from 3 independent studies and following mouse euthanasia and lung perfusion, lungs were inflated with 1 ml digestion solution/mouse containing 0.5 mg/ml Liberase$^{TM}$ (medium Thermolysin concentration; Millipore Sigma, St. Louis, MO) and 235.5 U/ml DNAse I in Hank's Balanced Salt Solution (pH = 7.2). Inflated lungs were dissociated with gentleMACS dissociator (Miltenyi Biotech, Auburn, CA) and incubated for 15 minutes at 37˚C in a shaking incubator. Digestion solution activity was neutralized with FA3 buffer (10mM HEPES, 2mM EDTA, 1% FBS in PBS). The single cell lung suspensions were incubated with CD16/32 (Fc Block, Cat#101320, Lot#B276722) Biolegend, San Diego, CA) to minimize nonspecific antibody staining. Next, cells stained with mAbs directed against rat anti-mouse CD45 (clone 30-F11, Cat#563053, Lot#8330943), Ly6C (clone AL-21, Cat#560596, Lot#9267106), Ly6G (clone 1A8, Cat#551461, Lot#21331), CD11b (clone M1/70, Cat#550993, Lot#8232762), and hamster anti-mouse CD11c (clone N418, Cat#61-0114-82, Lot#2133313, Invitrogen, Eugene, OR), and live/dead fixable blue dead cell stain kit (Invitrogen, Eugene, OR). Antibodies to CD11c from Invitrogen, and the remainder from BD Biosciences (San Jose, CA). Flow-sorting was done with FACSAria II (BD Biosciences). Live CD45$^+$ singlets were gated on Ly6C$^+$Ly6G$^+$ to sort neutrophils. Lymphocytes (based on FSC and SSC) and neutrophils (based on Ly6C and Ly6G staining) were then reverse gated to further select for 3 monocyte/macrophage populations: macrophage (CD11c$^{high}$, CD11b$^{variable}$), monocytes-macrophages (CD11c$^{intermediate}$,CD11b$^{high}$), and monocytes (CD11c$^-$, CD11b$^{high}$). This gating strategy for monocyte/macrophage populations is consistent with previous reports by us and others [17–20].

## RNA isolation

The 4 cell-sorted populations were counted, assessed for viability by trypan blue exclusion (>95%), washed and lysed with RLT buffer containing β-mercaptoethanol for RNA isolation as per manufacturer's instructions with Qiagen RNAeasy Micro Kit (Qiagen, Germantown, MD).

## NanoString nCounter system

Quality and quantity of total RNA was evaluated using a Fragment Analyzer (Agilent, Santa Clara, CA) and Nanodrop (ThermoFisher), respectively. Total RNA (25–50 ng) was hybridized and processed per the manufacturer's suggested protocol with capture and reporter probes to prepare target-probe complexes using reagents from the Mouse Autoimmune profiling panel containing 771 genes (NanoString, Seattle, WA). Complexes were purified, immobilized and aligned on a cartridge for counting on the nCounter system and processed as per the manufacturer's instructions.

For NanoString analyses, three independent studies of 2–3 pooled mice per group/experiment (N = 8 total mice/group) were analyzed by ANOVA with Tukey's multiple comparisons test. Arthritis inflammatory scores over time was also analyzed by ANOVA. Gene expression data were normalized to 20 housekeeping genes, treatment groups (CIA, ODE and CIA +ODE) were compared to Sham, and data plotted as fold-change. ANOVA with Tukey's multiple comparison test was used on myeloid-derived suppressor cell (MDSC) posthoc analysis. Bar graphs were used to depict means with standard errors of the ratio change in MDSCs normalized to Sham (percentile of MDSC treatment group divided by percentile of MDSC Sham group). Statistical analyses were performed using the GraphPad Prism software, version 8.4.3 (GraphPad, San Diego, CA), and statistical significance accepted at p-values <0.05.

## Results

### Agriculture (swine) exposure-related ODE-induced airway inflammation coupled with arthritis modeling

Consistent with the previously published study describing this model system [11], the highest arthritis inflammatory scores were demonstrated for CIA+ODE at 5 weeks (Fig 1A). Throughout the 5 weeks, arthritis inflammatory scores were recorded weekly, and as early as 1 week, there were increases in arthritis inflammatory scores in CIA+ODE and CIA alone as compared with Sham. At 4 weeks, there were significant increases in ODE alone. Previous lung pathology investigations [11] reported increases in alveolar and bronchiolar compartment inflammation and increases in cellular aggregates with ODE and CIA+ODE as compared to Sham, but the number and size of cellular aggregates were reduced in CIA+ODE as compared to ODE alone. Instead, pro-lung fibrosis mediators including lung hyaluronan and fibronectin were increased in CIA+ODE as compared to ODE alone [11]. In these current studies, lung sections from the previous study were stained with H&E from each treatment group showing increased inflammation in CIA+ODE (Fig 1B). Sections were also stained with markers to identify macrophages (CD68), neutrophils (MPO), T cells (CD3), and B cells (CD45R) (Fig 1C and 1D) distributed in the parenchyma, peribronchiolar and perivascular region. Infiltration of recruited CCR2+ inflammatory monocytes was also demonstrated in the CIA and ODE groups, and these cells were further potentiated in the CIA+ODE group (Fig 1E and 1F).

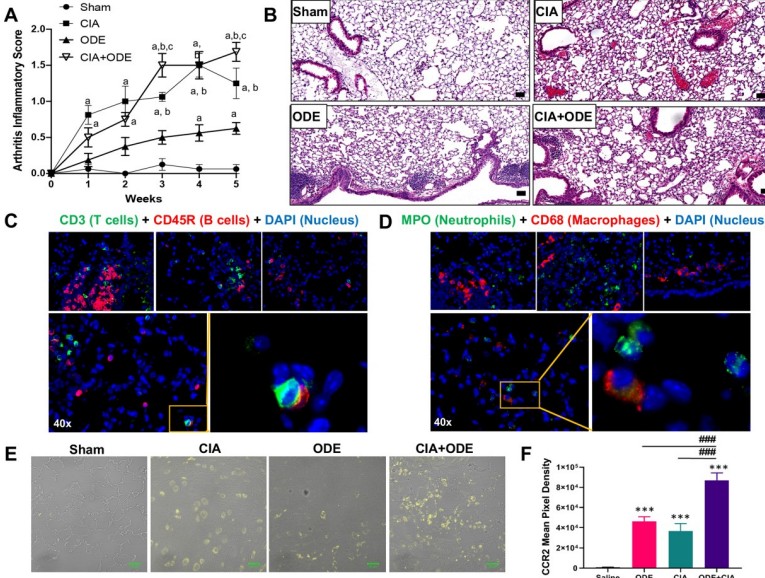

**Fig 1. Agriculture (swine) exposure-related Organic Dust Extract (ODE) induced airway inflammation coupled with Collagen Induced Arthritis (CIA) model.** (**A**) Line graph depicts mean with SE bars of arthritis inflammatory score at respective time points from treatment groups. Statistical difference versus sham denoted as "a" ($p < 0.05$); versus ODE denoted "b" ($p < 0.05$); versus CIA denoted as "c" ($p < 0.05$) as determined by two-way ANOVA. N = 8 mice/group from 3 independent studies. (**B**) Representative H&E-stained lung section from each treatment group at 10 X magnification with line scale (100 pixels). Representative images of lungs of CIA+ODE at 40x (parenchyma and peribronchiolar/perivascular regions), immunostained with CD3 (T cells) and CD45R (B cells) (**C**), and myeloperoxidase (MPO, neutrophils) and CD68 (macrophages) (**D**). Zoomed images highlight B-cell-T-cell interaction (**C**) and macrophages with neutrophils (**D**). CCR2+ inflammatory monocytes were increased with ODE and CIA treatment conditions in murine lungs. (**E**) Representative image of CCR2 expression (yellow) of lung tissue from each treatment group. (**F**) Bar graph depicts mean with standard error bars of CCR2 staining (N = 5/group). Statistical difference ***$p < 0.001$ vs. saline/sham control and ###$p < 0.001$ denoted by line between groups.

These studies lay the foundation for current studies delineating cellular and gene determinations through high-throughput analysis.

## ScRNA-seq identifies 14 unique immune cell subsets

The 10x genomics platform was utilized to cumulatively capture all lung cells. In total, 16,822 cells were analyzed with a mean of 42,901 post-normalization reads per cell and 956 median genes per cell. Unsupervised clustering was performed on 11,577 CD45$^+$ cells and plotted on *t*-distributed Stochastic Neighbor Embedding (t-SNE). Projection of cells was colored based on unique molecular identifier (UMI) count to identify level of transcripts among the cells. The average UMI count range was roughly between 2,000 to 12,000. Cells that were distributed in the middle showed the highest level of transcripts while cells at the top showed the lowest level (Fig 2A). Unsupervised clustering on the t-SNE projected cells revealed 14 unique immune cell subsets coded by different colors and arbitrary numbers (Fig 2B). Clusters 3, 4 and 8 were identified as neutrophil subsets based on distribution of *Csf3r* (granulocyte colony stimulating factor receptor [21]) in t-SNE analysis (Fig 2C). Macrophages were distributed in the middle and were identified with *Cd11c* (ITGAX) expression in clusters 1, 2, 5, 11, 14, and partially in cluster 10. Monocytes (inflammatory monocytes) were identified with *F13a1* expression in cluster 12. Cluster 9 showed high levels of *Ccl5* expression suggesting the presence of NK cells. Similarly, *Cd19* expressing cells in cluster 7 identified B lymphocytes, and *Trbc2* expression in clusters 6 and 13 identified T lymphocytes, along with expression in the NK cell population (cluster 9). Dendritic cells (DCs) were located in cluster 10 and were characterized by *Siglech* expression, particularly evident in cluster 10a (Fig 2C).

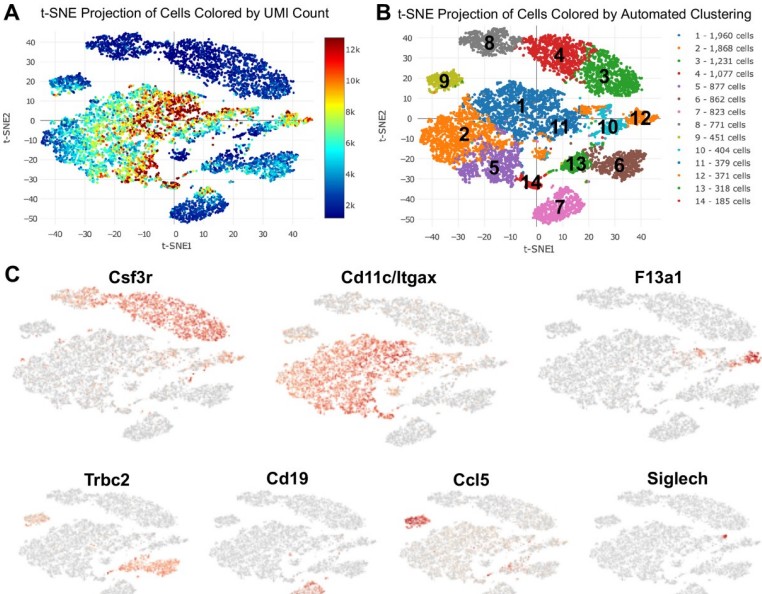

**Fig 2. Unsupervised single-cell transcriptional profiling of lung CD45$^+$ cells identifies 14 unique clusters among Sham, CIA, ODE, and CIA+ODE treatment groups.** Lung immune cells were isolated from mice (N = 2 mice/group) treated with Sham, CIA, ODE and CIA+ODE. (**A**) T-distributed stochastic neighbor embedding (*t-SNE*) plot shows projection of unique molecular identifier (UMI) count among cell clusters. (**B**) Distribution of cells by unsupervised clustering in *t-SNE* showing lung immune cell populations. (**C**) Major lung cell types identified by signature genes including *Csf3r* (neutrophils), *Cd11c/Itgax* (macrophages), *F13a1* (monocytes), *Trbc2* (T lymphocytes), *Cd19* (B lymphocytes), *Ccl5* (NK cells), and *Siglech* (dendritic cells).

## CIA and ODE drive unique distributions of immune cells within identified clusters

The 4 treatment groups (Sham, CIA, ODE, and CIA+ODE) exhibited unique distributions of lung immune cells among the identified clusters (Fig 3A and 3B). Among the neutrophil clusters, Sham was exclusively represented by cluster 4, but not cluster 3 or 8. In contrast, the CIA group almost entirely showed neutrophil distribution in cluster 3. The ODE group demonstrated selective distribution of neutrophils in cluster 8 with overlap into cluster 4. In the combination exposure CIA+ODE group, there was broader distribution of neutrophils with predominance in cluster 3, but also evidence for distribution in cluster 4 and partially in cluster 8 (Fig 3A and 3B).

Based on the shifts observed in the cell populations among treatment groups, a manual subclustering was performed to delineate exact number of cells distributed among the sub-clusters (Fig 3C). Among the macrophage clusters, the ODE group had prevalence in clusters 5, 1b and 1c compared to the CIA group. Likewise, the ODE group lacked clusters 1a and 2a. A subset of cluster 12 (12b) and cluster 10 (10b) were [22] unique to the CIA group, while clusters 10a and 12a were unique to the ODE group (Fig 3). The combination group with CIA+ODE showed a mixed population representing CIA and ODE, while leaning more towards the CIA group (Fig 3).

Lymphocyte populations were confined to clusters 6, 7, 9, and 13, and were represented in all treatment groups, although modest shifts in cell population distribution were observed. Particularly, NK cells (cluster 9) and B cells (cluster 7) were differentially expressed in ODE and CIA treatment groups with apparent shifts from cluster 9a in ODE to cluster 9b in CIA and shifts from cluster 7a in ODE to cluster 7b in CIA, respectively (Fig 3). Similar to the macrophage clusters, the CIA+ODE group portrayed CIA and ODE group while inclining more towards the CIA group (Fig 3).

## Three distinct neutrophil populations revealed by scRNA-seq among treatment groups

Unsupervised clustering segregated 3 populations of granulocytes/neutrophils that were marked by unique gene expression (Fig 4A). Relative gene-expression compared to all other cell populations as log2 fold-change was plotted in a heat map (Fig 4B) and violin plot (Fig 4C) to compare transcript levels as well as cell distributions at different expression levels. Cluster 8 showed increased expression of inflammatory genes such as *Ccl3*, *Ccl4*, *Cxcl2*, *Upp1* and *Marcks1* (log2 fold-change range: 4.98–6.35), which are genes commonly upregulated in

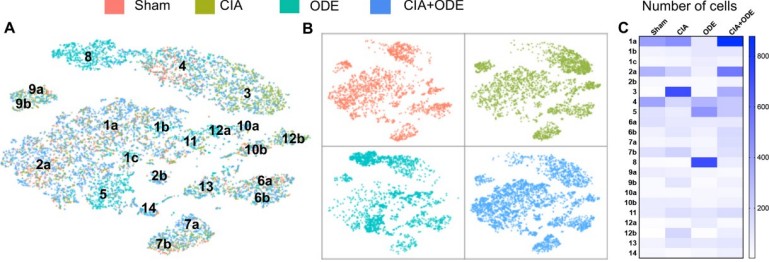

**Fig 3. Distribution of aggregated gene clusters among treatment groups.** Sham, CIA, ODE and CIA+ODE treatment groups demonstrate differences in cell distribution among 14 gene clusters. Sham is represented by red, CIA by green, ODE by teal, CIA+ODE by blue. (**A**) tSNE plot with all the treatment groups merged with respective colors (**B**) Treatment groups are plotted individually with their respective colors to show cell distribution. (**C**) Heatmap shows number of cells per sub-cluster among treatment groups by manual sub-clustering (total cells = 11,577).

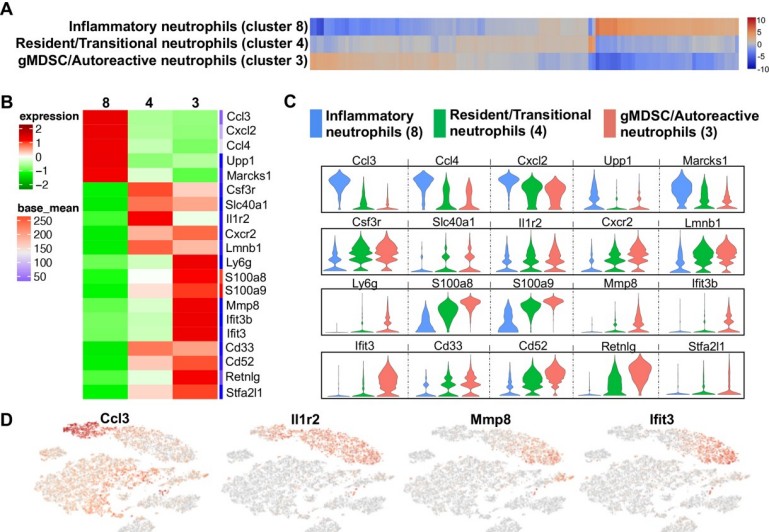

**Fig 4. Neutrophil populations segregated by unsupervised clustering relate to cell-programming among treatment groups.** (**A**) Heatmap shows the top 120/N upregulated genes for 3 distinct neutrophil clusters ranked by log2 fold-change, where N = total number of clusters. (**B**) Top 5–10 genes of the three neutrophil clusters plotted in heatmap to show differences and gene names in transcript levels. (**C**) Violin plots show expression with population distribution among inflammatory neutrophils, represented by blue (cluster 8), resident/transitional neutrophils, represented by green (cluster 4) and granulocytic myeloid-derived suppressor cells (gMDSC)/autoreactive neutrophils, represented by red (cluster 3). The y-axis indicates normalized expression value, log2 (average UMI count + 1). (**D**) Representative gene from each neutrophil cluster showing their distribution in t-SNE.

activated neutrophils (Fig 4B and 4C, and S1 Table) [23–27]. Moreover, cluster 8 was exclusive to the ODE group. In contrast, cluster 3 exhibited increased expression of genes associated with immunosuppression and autoreactivity as well as genes that are characteristic of granulocytic MDSC (gMDSC) such as *S100a8*, *S100a9*, *Mmp8*, *Ifit3b*, *Ifit3*, *Cd33*, *Cd52* and *Stfa2l1* (log2 fold-change range: 5.16–6.03) (Fig 4B and 4C, and S1 Table) [28–34]. Based on the gene expression profiles, cluster 8 was identified as "inflammatory neutrophils" and cluster 3 was identified as "gMDSC/autoreactive neutrophils". Intermediate to the two cell subsets, another neutrophil sub-set (cluster 4) was identified as resident/transitional neutrophils. This population of neutrophils (Fig 4B and 4C, and S1 Table) demonstrated increased expression of *Csf3r*, *Il1r2*, *Slc40a1*, *Cxcr2*, and *Lmnb1* genes (log2 fold-change range: 3.91–4.24) that are required in neutrophil differentia-tion and trafficking [21, 35–40]. Neutrophils in the CIA+ODE group was distributed more like the CIA than the ODE group with predominant segregation in clusters 3 and 4 (Fig 3).

Signature genes were selected to highlight respective neutrophil populations on the t-SNE plot (Fig 4D). *Ccl3* was selected to highlight "inflammatory neutrophil" as *Ccl3* enhances recruitment and activation of neutrophils in a paracrine fashion [23, 24]. Because *Il1r2* gene encodes for type 2 interleukin-1 receptor and is constitutively expressed in mouse neutrophils [35], it identified all subsets of neutrophils in the t-SNE clusters (Fig 4D). Autoreactive neutro-phils/gMDSCs were exclusively positive for *Mmp8* and *Ifit3* in t-SNE. *Mmp8* is a neutrophil col-lagenase [41, 42] and *Ifit3* codes for interferon induced protein with tetratricopeptide repeats 3, as both are highly upregulated in gMDSCs and can suppress immune response [31, 32].

## Identification of unique macrophage/monocyte/DC populations

Based on gene expression patterns among segregated populations found with unsupervised clustering, 5 discrete macrophage clusters, 1 inflammatory monocyte cluster, and 1 DC cluster

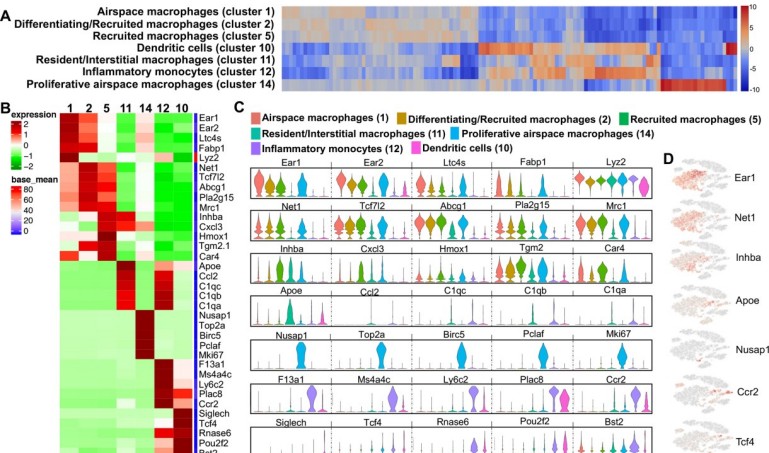

**Fig 5. Differences in transcript levels among monocyte-macrophages and dendritic cell (DC) populations.** (**A**) Heatmap shows the top 120/N upregulated genes for 7 macrophages/monocytes and DC clusters ranked by log2 fold-change, where N = total number of clusters. (**B**) Heatmap showing differences among top 5 genes with gene names of the macrophages/monocytes and DC populations. (**C**) Expression levels with violin plots among airspace macrophages (red, cluster 1), differentiating/recruited macrophages (brown, cluster 2), recruited macrophages (green, cluster 5), resident/interstitial macrophages (teal, cluster 11), proliferative airspace macrophages (light blue, cluster 14), inflammatory monocytes (lavender, cluster 12), dendritic cells (pink, cluster 10). The y-axis indicates normalized expression value, log2 (average UMI count + 1). (**D**) Distribution of respective cell populations by a representative gene in t-SNE plot.

were identified (Fig 5A). Cluster 1 was termed "airspace macrophages" based on increased expression (log2 fold-change range: 1.52–1.98) of *Ear1*, *Ear2*, *Ltc4s*, *Fabp1* and *Lyz2* compared to other clusters (Fig 5B–5D), representing genes responsible for metabolism and inflammation/resolution [43–47] (S2 Table). Cluster 2 was labeled as "differentiating/recruited macrophages" as this cluster exhibited the highest expression (log2 fold-change range: 1.27–1.54) of *Net1*, *Tcf7l2*, *Abcg1*, *Pla2g15* and *Mrc1* representing genes implicated in differentiation, antigen uptake, and macrophage recruitment [48–52] (Fig 5B–5D). Cluster 2 macrophages expressed genes associated with alternatively activated macrophages (M2 macrophages) [49, 52], and upregulate pathways for lipoprotein metabolism and redox signaling (S2 Table). Cluster 5 was identified as "recruited macrophages" based on the disproportionate expression (log2 fold-change range: 1.5–1.96) of *Inhba*, *Cxcl3*, *Hmox1*, *Tgm2*, and *Car4* (Fig 5B–5D). Cluster 5 was heterogeneous with *Inhba* representing classically activated (M1) macrophages or *Hmox1* as M2 macrophages, and also included genes that are involved in inflammation, adipogenesis, homeostasis and phagocytosis [53–62] (S2 Table).

Sham, CIA and CIA+ODE groups showed similar distribution of airspace (particularly cluster 1a) and recruited macrophages (especially cluster 2a) while the ODE group had a substantial reduction in these macrophage populations with segregation towards the center of the t-SNE plot in clusters 1b and 1c (Fig 3).

Cluster 11 was designated as "resident interstitial macrophages" with high transcript levels of *Apoe*, *Ccl2* and complement genes such as *C1qc*, *C1qb* and *C1qa* (Fig 5B–5D) ranging from log2 fold-change of 3.54–3.91. This population also displayed heterogeneity with expression of both M1 and M2 genes [47, 63–67] involved in inflammation and resolution (S2 Table). Resident interstitial macrophages were more evident in the ODE compared to other treatment groups (Fig 3).

As reported by Mould *et. al.* [68], we also identified a distinct cluster of macrophages (cluster 14) with very high expression of proliferative and mitotic genes (log2 fold-change range:

5.79–8.12) including *Nusap1*, *Top2a*, *Birc5*, *Pclaf*, and *Mki67* (Fig 5B–5D), which were termed "proliferative airspace macrophages." The upregulated pathways included cell-cycle, mitosis or proliferation-related pathways (S2 Table) [68–70]. These proliferative airspace macrophages were represented largely by the ODE and CIA+ODE groups (Fig 3).

Cluster 12 represented a unique cell population identified as "inflammatory monocytes." This population exhibited increased expression of *F13a1*, *Ms4a4c*, *Ly6c2*, *Plac8* and *Ccr2* (Fig 5B–5D) (log2 fold-change range: 4.65–6.53), all characteristically expressed in inflammatory monocytes and often correlated with anti-viral and/or autoimmune responses [71–79]. Although CIA+ODE group had a pronounced cluster 12, 12b represented the CIA group and ODE group exhibited cluster 12a (Fig 3). A cell population in cluster 10 was identified demonstrating elevated expression in genes (log2 fold-change range: 2.61–6.48) of *Siglech*, *Tcf4*, *Rnase6*, *Pou2f2* and *Bst2* (Fig 5B–5D), which are distinctive of DCs [80–86]. This population demonstrated characteristics of plasmacytoid DCs involving genes associated with innate immunity and anti-inflammatory pathways. The DC predominated with ODE and CIA+ODE groups in cluster 10a, whereas cluster 10b predominated with Sham (Fig 3). Overall, CIA +ODE group distribution of monocyte-macrophages followed neutrophils with overrepresentation of CIA group (Fig 3).

## Lymphocytes segregate in four clusters among treatment groups

Four discrete lymphocyte clusters were found in the analysis (Fig 6A). Cluster 6 was identified as "T lymphocytes", with increased expression (log2 fold-change range: 4.87–7.04) of *Lef1*, *Igfbp4*, *Tcf7*, *Cd3d*, and *Cd3e* (Fig 6B–6D). This population favored type 2 CD4$^+$ cells based upon the expression of differentiating or expanding population of T lymphocytic genes [87–91]. Cluster 6a was more represented by Sham whereas cluster 6b was represented by CIA and CIA+ODE treatment groups. The ODE group had sparse distribution between cluster 6a and 6b (Fig 3). In contrast to cluster 6, cluster 13 exhibited increased expression of genes indicative of activated T lymphocytes including *Icos*, *Thy1*, *Cd3g*, *Ikzf2* and *Maf* (log2 fold-change range: 4.87–5.87) and thus were termed as "effector T lymphocytes" (Fig 6B–6D) with upregulation

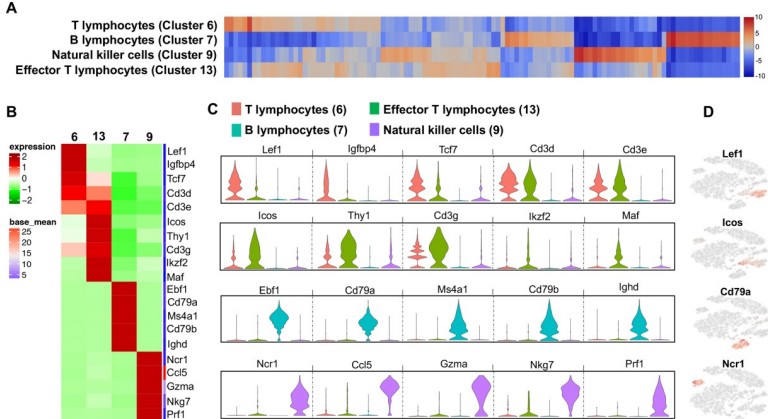

**Fig 6. Four discrete lymphocyte populations suggest heterogeneity among treatment groups.** (**A**) Heatmap shows the top 120/N upregulated genes for 5 distinct lymphocyte clusters ranked by log2 fold-change, where N = total number of clusters. (**B**) The top 5 genes with gene names of each lymphocyte cluster are shown in heatmap. (**C**) Violin plots show variability in transcript levels among cell populations including T lymphocytes (red, cluster 6), effector T lymphocytes (green, cluster 13), B lymphocytes (teal, cluster 7) and natural killer cells (lavender, cluster 9). The y-axis indicates normalized expression value, log2 (average UMI count + 1). (**D**) Representative genes from each cluster show their distribution in t-SNE plot.

of co-stimulatory and adaptive immune pathways. Subtle differences were observed in the distribution of activated T lymphocytes among the treatment groups (Fig 3).

Cluster 7 was remarkable for increased gene expression characteristic of B lymphocytes such as *Ebf1*, *Cd79a*, *Ms4a1*, *Cd79b* and *Ighd* (log2 fold-change range: 7.29–7.77) (Fig 6B–6D). Along with genes implicated in B-cell differentiation, memory, signaling and autoimmunity, this cluster showed striking similarities with upregulated pathways in the T lymphocyte population (S3 Table) [92–95]. B lymphocytes had an overall distribution in CIA+ODE group, but largely represented as cluster 7b in the CIA group. The ODE group had very few B lymphocytes with sparse distribution (Fig 3).

NK cell-specific gene expression was increased in cluster 9 with *Ncr1*, *Ccl5*, *Gzma*, *Nkg7* and *Prf1* (Fig 6B–6D) ranging from 7.83 to 7.87 log2 fold-change (S3 Table). These genes and pathways were predominately related to NK cell recruitment, activation and effector function [96–101]. The Sham and ODE groups were represented by cluster 9a while the CIA group had more of cluster 9b. The CIA+ODE group had 9a and 9b clusters (Fig 3).

## Differential gene expression of ex vivo sorted lung neutrophils across treatment groups represent disease progression

To understand the relevance of myeloid-derived lung cells in RA and RA-associated lung disease, these studies sought to determine whether gene expression of sorted lung myeloid-derived cells corresponded to disease-specific findings among treatment groups. Lung neutrophils were isolated by fluorescence activated cell sorting (FACS) based on traditional cell surface markers as percent of CD45$^+$ cells that were Ly6C$^+$ Ly6G$^{high}$ (Fig 7A and S1 Fig). Sham represented resident neutrophils in the lungs at baseline [102]. By NanoString analysis, upregulated genes of isolated neutrophils resembled the gene expression demonstrated in scRNA-seq data by respective treatment groups. Neutrophils isolated from lungs of CIA and CIA +ODE groups (as compared to Sham) demonstrated increased transcript levels of genes

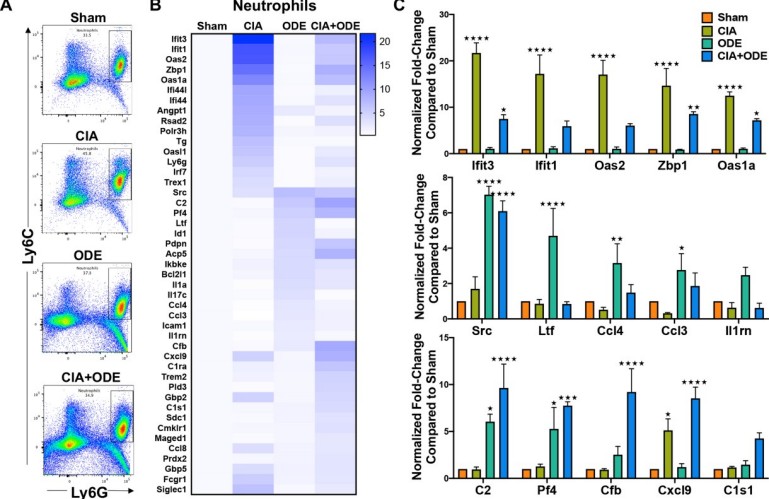

**Fig 7. Treatment group-specific gene expression pattern demonstrated in isolated lung neutrophils.** Neutrophils were sorted from lung digests as live, singlets, CD45$^+$, non-lymphocytes, Ly6C$^+$ and Ly6G$^+$. (**A**) Representative dot plots of Ly6C$^+$ Ly6G$^+$ neutrophils sorted from Sham, CIA, ODE and CIA+ODE treatment groups shown. (**B**) Heat map of fold-change of top 15 genes normalized to 20 housekeeping genes from each treatment group compared to Sham. (**C**) Bar graphs reflect the mean of normalized fold-change with standard error bars of representative genes from each treatment group Sham (red), CIA (green), ODE (teal), and CIA+ODE (blue). N = 3 (3 independent experiments with 2–3 mice pooled, 8 total mice). *P<0.05, **P<0.01, ***P<0.001, ****P<0.0001.

involved in autoimmunity as well as genes associated with gMDSCs/autoreactive neutrophils. These included (CIA and CIA+ODE): *Ifit3* (21.7 and 7.49-fold), *Ifit1* (17.2 and 5.9-fold), *Oas2* (17.0 and 6.0-fold), *Zbp1* (14.6 and 8.5-fold), *Cxcl9* (5.1 and 8.5-fold), and *Oas1a* (12.5 and 7.2-fold) (Fig 7B and 7C). Interestingly, the CIA+ODE group also showed gene expression that paralleled that of the ODE group, including increased transcript levels of *Src* (CIA+ODE: 6.1-fold and ODE: 7-fold), *Pf4* (7.7 and 5.3-fold), and complement cascade genes such as *C2* (9.6 and 6-fold-change), and *Cfb* (9.2 and 2.5-fold-change) (Fig 7B and 7C). In contrast, the ODE group demonstrated exclusive upregulation of *Ltf* (4.7-fold-change), *Ccl4* (3.2-fold), *Ccl3* (2.8-fold) and *Il1rn* (2.5-fold) as compared to Sham, consistent with the inflammatory neutrophil cluster (Fig 7B and 7C). There was a single complement cascade gene (*C1s1*) that was exclusively upregulated in CIA+ODE (4.3-fold) (Fig 7C).

## Macrophage and monocyte populations from CIA and CIA+ODE groups exhibit gene profiles comparable to RA and RA-associated lung disease, respectively

After excluding dead cells, doublets, lymphocytes and neutrophils (Ly6G⁻), 3 separate lung monocyte-macrophage populations were sorted based upon CD11c and CD11b expression (Fig 8A and S1 Fig) [17, 18]. These 3 populations are 1) CD11c$^{high}$CD11b$^{variable}$ macrophages, 2) CD11c$^{intermediate}$CD11b$^{high}$ mono-macs and 3) CD11b$^{high}$CD11c$^{-}$ monocytes. Of the CD11c$^{high}$ macrophages, the expression of CD11b (evident on the pseudocolor plots; Fig 8A) shifts to the right: CIA+ODE>ODE>CIA as compared to Sham. Increasing expression of CD11b on CD11c+ macrophages suggest an activated phenotype [18, 103]. Infiltrating inflammatory monocytes and recruited monocytes/macrophages have been implicated in other experimental lung fibrotic experimental settings [104–106]. Similar to neutrophils, the isolated

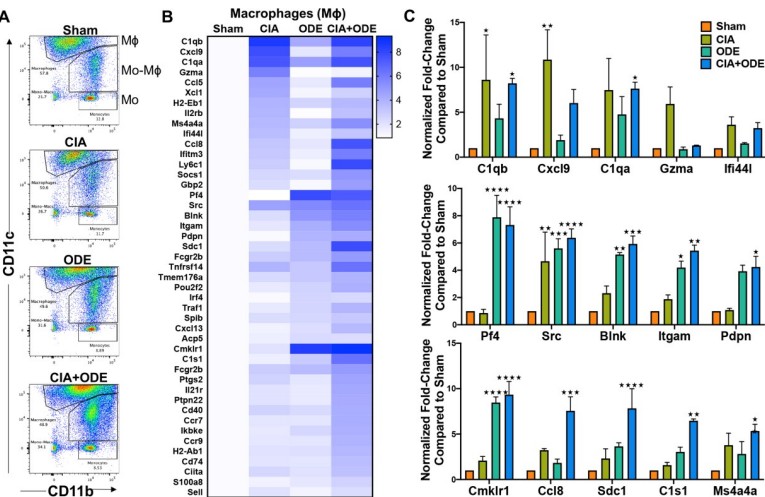

**Fig 8. Treatment group-specific gene expression pattern demonstrated in isolated macrophages.** Three monocyte/macrophage populations were sorted from lung digests as live, singlets, CD45$^{+}$, non-lymphocytes, Ly6C$^{-}$ and Ly6G$^{-}$, and identified as separate populations with variable expression of CD11b and CD11c. RNA was isolated from these populations and subjected to NanoString nCounter analysis. (**A**) Representative dot plots of the populations sorted as: (1) macrophages (CD11c$^{high}$, CD11b$^{variable}$), (2) monocytes-macrophages (CD11c$^{intermediate}$,CD11b$^{high}$), and (3) monocytes (CD11c$^{-}$, CD11b$^{high}$) from each treatment of Sham, CIA, ODE, and CIA+ODE shown. (**B**) Heat map of fold-change of top 15 genes normalized to 20 housekeeping genes from each treatment group compared to Sham. (**C**) Bar graphs depict mean with standard error bars of representative genes from each treatment group Sham (red), CIA (green), ODE (teal), and CIA+ODE (blue). N = 3 (3 independent experiments with 2–3 mice pooled), 8 total mice. *P<0.05, **P<0.01, ***P<0.001, ****P<0.0001.

macrophage population also demonstrated increased gene expression in CIA and CIA+ODE group that included *C1qb* (8.6 and 8.2-fold), *Cxcl9* (10.9 and 6-fold), *C1qa* (7.5 and 7.6-fold), *Ifi44l* (3.6 and 3.2-fold), *Cmklr1* (2.1 and 9.3-fold), *Ccl8* (3.2 and 7.6-fold), *Sdc1* (2.3 and 7.8-fold), and *Ms4a4a* (3.8 and 5.3-fold). Increased expression of *Gzma* was unique to the CIA group (6-fold) (Fig 8B and 8C), while *C1s1* was upregulated in both CIA+ODE (6.5-fold) and ODE (3-fold) groups along with *Pf4* (7.9 and 7.3-fold), *Itgam* (4.2 and 5.4-fold), and *Pdpn* (4 and 4.2-fold). Expression of *Src* (4.7, 5.6 and 6.4-fold) and *Blnk* (2.3, 5.1 and 5.9-fold) was increased in all treatment groups (CIA, ODE and CIA+ODE) as compared to Sham (Fig 8B and 8C).

The CD11c$^{intermediate}$CD11b$^+$ monocyte-macrophage population demonstrated increased expression of several interferon-associated and other genes implicated in autoimmune responses in CIA and CIA+ODE groups including *Gbp2* (7.4 and 6.9-fold), *Zbp1* (7 and 2-fold), *Ifi44* (6.6 and 4.3-fold), *Ifi44l* (6 and 5.1-fold), *Cxcl9* (5.8 and 8.1-fold), and *Fcgr1* (2.4 and 2.9-fold). ODE and CIA+ODE groups demonstrated increased expression of *Cxcl5* (3.4 and 2.6-fold), *Pdpn* (4.2 and 3.3-fold), *Pf4* (5.4 and 3.8-fold), and *Cxcl13* (6.8 and 3.6-fold) compared to Sham (Fig 9A and 9B). *Cfb* (5.9, 4.1, and 16.3-fold), *Ccl8* (11.3, 5.1, and 6.7-fold), and *C1s1* (5.8, 7, and 19.3-fold) were overexpressed in CIA, ODE and CIA+ODE groups, respectively, compared to Sham. Expression of non-canonical I-kappa-B kinase, *Ikbke*, associated with anti-viral responses and autoimmune diseases, was increased in the CIA+ODE group (2.5-fold) compared to Sham (Fig 9A and 9B).

The monocyte population (CD11c$^-$CD11b$^+$) was unique because all the upregulated genes including *Oasl1* (15 and 4.6-fold), *Oas1a* (15.6 and 15.3-fold), *Oas2* (11.7 and 12.4-fold), *Ifi44* (11.0 and 9.5-fold), *Ifi44l* (11.1 and 11.1-fold), *Siglec1* (11.1 and 6.1-fold), *Gbp2* (2.9 and 5.3-fold), *Gbp5* (3.8 and 4.3-fold), *Stat1* (3.3 and 3.8-fold), and *Isg15* (4.1 and 3.7-fold) were increased in both CIA and CIA+ODE groups respectively (Fig 10A and 10B). Moreover, these

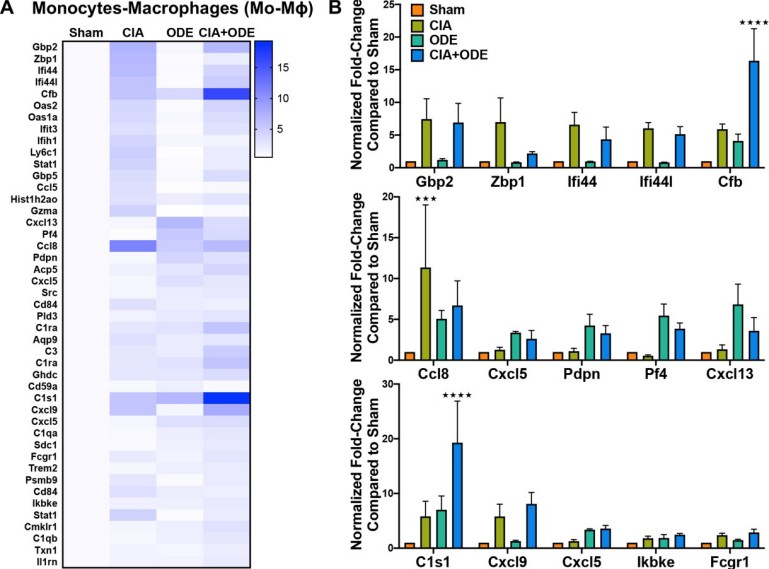

**Fig 9. Treatment group-specific gene expression pattern demonstrated is isolated monocyte-macrophage population.** Monocytes-macrophages were sorted as CD11c$^{intermediate}$,CD11b$^{high}$. (**A**) Heat map of fold-change of top 15 genes/treatment group (CIA, ODE and CIA+ODE) normalized to 20 housekeeping genes compared to Sham. (**B**) Bar graphs of mean with standard error bars of representative genes from each treatment group Sham (red), CIA (green), ODE (teal), and CIA+ODE (blue). N = 3 (3 independent experiments with 2–3 mice pooled, 8 total mice). ***P<0.001, ****P<0.0001.

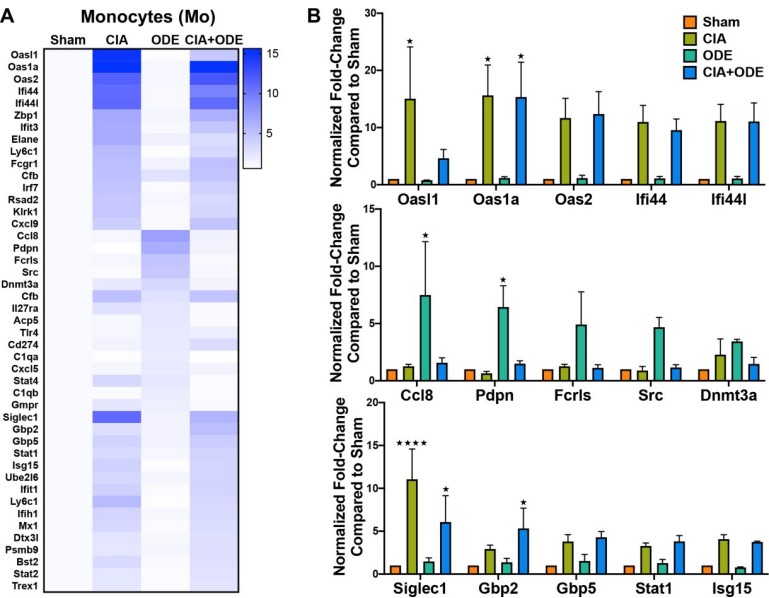

**Fig 10. Treatment group-specific gene expression pattern.** Demonstrated in isolated monocytes. Monocytes were sorted as CD11c⁻, CD11b^high. (**A**) Heat map of fold-change of top 15 genes/treatment group (CIA, ODE and CIA +ODE) normalized to 20 housekeeping genes compared to Sham. (**B**) Bar graphs depict mean with standard error bars of representative genes from each treatment group Sham (red), CIA (green), ODE (teal), and CIA+ODE (blue). N = 3 (3 independent experiments with 2–3 mice pooled, 8 total mice). $*P<0.05$, $****P<0.0001$.

genes are mostly associated with autoimmunity or immunosuppression [32, 57]. The ODE group exhibited higher transcript levels of *Ccl8* (7.5-fold), *Pdpn* (6.4-fold), *Fcrls* (4.9-fold) and *Src* (4.7-fold), consistent with the other sorted neutrophil and monocytes/macrophages populations. While *Cfb* expression was increased in all 3 treatment groups, *Dnmt3a* was upregulated (2.3 and 3.4-fold) in ODE and CIA+ODE groups, respectively (Fig 10A and 10B), but not in CIA.

Finally, gMDSCs were identified as Ly6C⁺Ly6G^high CD11b^high SSC^high (S2 Fig) [107, 108] on posthoc gating of sorted neutrophil populations resulting in non-significant variations across the treatment groups (Fig 11A). In contrast, mMDSC defined as Ly6G⁻ CD11b⁺ Ly6C^high SSC^low cells (S2 Fig) [107, 108] were increased with CIA but decreased with ODE and CIA +ODE (Fig 11B).

## Discussion

In this study, scRNA-seq analysis was applied to whole lung immune cells from a mouse model of RA-associated inflammatory lung disease with key findings confirmed in sorted lung cell populations and NanoString analysis. Building upon the preclinical model of RA-associated inflammatory lung disease [11], we report a number of key findings in this study including: (a) identification of 3 unique neutrophil populations including inflammatory, transient and immunosuppressive/autoreactive granulocytes among experimental groups, (b) heterogeneity among 5 macrophage populations including metabolically active, proliferative, differentiating, recruited, and residential with classical (M1) and alternatively (M2)-activated genes, (c) identification of 2 stages of T-lymphocytes (differentiating and effector), a B-cell population and a NK cell cluster, (d) variability in the distribution of cellular clusters among the treatment groups representing RA and RA-associated lung disease (CIA and CIA+ODE groups, respectively), (e) identification of gMDSC and mMDSC populations based on cell surface markers,

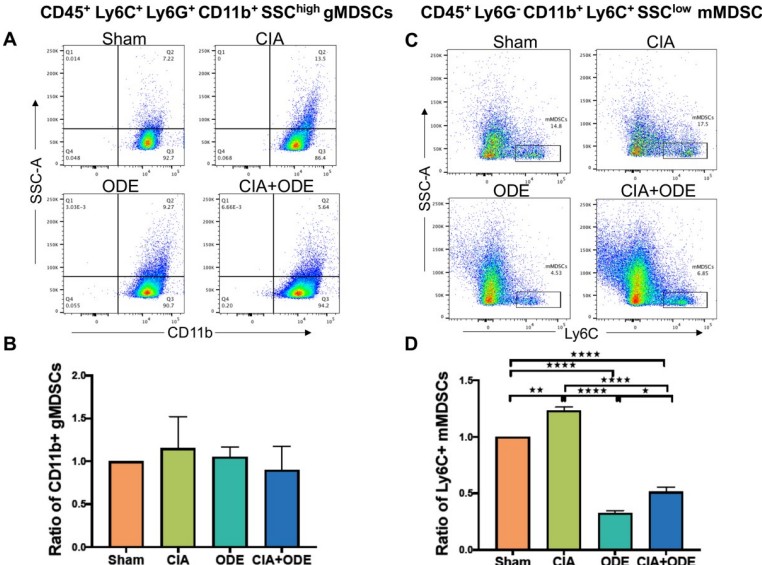

**Fig 11. Differences in myeloid-derived suppressor cells (MDSCs) identified in posthoc analysis among treatment groups.** (**A**) Dot plots show differences in CD45$^+$ Ly6C$^+$ Ly6G$^+$ CD11b$^+$ SSC$^{high}$ granulocytic (g) MDSCs among treatment groups. (**B**) Bar graph depicts mean with standard error bars of the ratio change in gMDSCs as normalized to Sham (percentile of gMDSC treatment group divided by percentile of gMDSC Sham group). (**C**) Dot plots show differences in CD45$^+$ Ly6G$^-$ CD11b$^+$ Ly6C$^+$ SSC$^{low}$ monocytic (m) MDSCs among treatment groups. (**D**) Bar graph depicts mean with standard error bars of the ratio change in mMDSCs as normalized to Sham (percentile of mMDSC treatment group divided by percentile of mMDSC Sham group). N = 3 (3 independent experiments with 2–3 mice pooled; 8 total mice). *P<0.05, **P<0.01, ****P<0.0001.

and (f) identification of unique genes (interferon-related/autoimmune and complement cascade) that are found in a mouse model of RA-associated inflammatory lung disease.

Occupational exposures from farming, construction, mechanics, medical and military waste have been associated with increased risk of development of RA and/or RA-associated lung disease [109–113]. However, precise mechanism(s) of the development of RA-associated lung disease with occupational and/or environmental inflammatory exposures is not known. Working towards identifying these mechanisms, previous studies demonstrated that repeated exposure to microbial-enriched ODE and particularly ODE+CIA increases citrullination and malondialdehyde-acetaldehyde (MAA)-adduction of lung proteins with a corresponding increase in circulating autoantibody concentrations, periarticular bone damage, and increased deposition of extracellular matrix proteins with a reduction in classical airway inflammatory markers [11, 114]. These findings suggested a transition of inflammatory lung disease towards a pro-fibrotic phenotype. Leveraging this co-exposure mouse model with CIA and ODE, several immune cell populations exhibiting unique gene expression signatures that were differentially distributed across treatment groups were demonstrated that suggest potential roles in the pathogenesis of RA, inflammatory lung disease, and inflammatory lung disease specific to RA.

Neutrophils have been classically related to inflammation and host response to pathogens [115]. Knowledge of their role in inflammation and homeostasis continues to evolve as various subsets of neutrophils have been identified and proposed based upon steady state, inflammatory, or anti-inflammatory programming [116]. Many intermediate phenotypes have also been defined, further complicating classifications and the proposed roles in both disease and homeostasis [117]. The 3 different neutrophil populations currently identified aptly signify an inflammatory (in ODE), anti-inflammatory (or autoreactive) (in CIA and CIA+ODE) and homeostatic (transient) (in Sham). In the cell sorting/NanoString studies, neutrophils were

identified and limited to Ly6C$^+$ Ly6G$^+$ cells based upon equipment limitations of simultaneous 4-cell capture ability. Future studies to include the Ly6G$^+$ Ly6C$^-$ cells that were predominately observed with ODE and CIA+ODE could also be informative. Further studies are needed to delineate precisely how these sub-populations program the lung immune response towards inflammatory and pro-fibrotic disease states.

MDSCs have been implicated in RA [118] as well as ILD [119]. However, their relationship to the RA-associated lung disease is not well-established. MDSCs are transient populations representing myeloid cells at various stages of differentiation that suppress immunity and are subdivided into granulocytic (g) or monocytic (m) origin [120–122]. MDSCs are identified by their high expression of *Nox2*, calprotectin (*S100a8/S100a9*), *Mmp8*, *Mmp9*, *Cd33*, and multiple interferon-inducible genes such as *Ifit3*, *Ifit1*, *Oas2*, *Zbp1*, *Ifi44*, *Ifi44l* and *Oas1a* [123, 124]. By scRNA-seq analysis, resolution of gMDSC was high with population segregation in cluster 3, which was driven by systemic arthritis induction (i.e. CIA and CIA+ODE groups). This finding was further strengthened by the RNA analysis of the sorted Ly6G$^+$ Ly6C$^+$ neutrophil population of corresponding treatment groups. Posthoc analysis confirmed that the sorted group contained gMDSCs with Ly6G$^+$ Ly6C$^+$ CD11b$^{high}$ SSC$^{high}$ gating [107, 108], but there was no difference across treatment groups. Unsupervised clustering of immune cells did not segregate mMDSCs. However, gene expression of sorted monocyte-macrophage populations based on CD11b and CD11c expression suggested the presence of mMDSC-like properties based upon the immunosuppressive genes that were elevated in CIA and CIA+ODE groups. Using a classical gating strategy (Ly6G$^-$ CD11b$^+$ Ly6C$^{high}$ SSC$^{low}$) for mMDSCs [108], mMDSCs were identified with FACS and found to be decreased in combination (CIA+ODE) exposure group. These findings are consistent with a recent report that showed that the expansion of MDSCs following tofacitinib treatment is inversely related to the progression of ILD in the SKG mouse model of RA-ILD [125]. These collective findings would suggest a potential protective role for lung MDSCs (particularly mMDSCs) in the development RA-related lung disease, and future studies are warranted to understand their role in disease manifestations to potentially develop novel targets for therapeutic interventions.

Macrophages are one of the most versatile immune cells with immense population heterogeneity and diverse functions [126, 127]. Macrophages are increasingly appreciated for their role in fibrosis, wound repair and resolution [47, 68, 128]. In this current study, the metabolically active airspace macrophages, resident, recruited and differentiating macrophages (clusters 1, 11, 5 and 2, respectively) contribute to inflammation and resolution, while the proliferative airspace macrophages (cluster 14) signify self-renewing properties to maintain a steady population in the lungs. These studies potentially open avenues for hypothesis generation based on various non-traditional genes (interferon-related/autoimmune and complement cascade) expressed in the macrophage subsets that have not been previously investigated in health and disease. The unique distribution of various macrophage clusters among the treatment groups, particularly clusters 1b, 1c, 5 and 11 in ODE group, and clusters 1a, 2a and 12b (inflammatory monocytes) in CIA and CIA+ODE groups signify their importance in disease transition from RA to RA-associated lung disease. Correspondingly, infiltrative CCR2$^+$ inflammatory monocytes were demonstrated with CIA and ODE with highest expression in combined exposures. Future studies could determine whether targeting this cell population results in a reduction of disease manifestations.

Subtle differences among lymphocyte populations (clusters 6, 7, 9 and 13) support earlier work demonstrating that B lymphocytes are skewed towards an autoreactive response following airborne biohazard exposure [114]. B lymphocytes have been recognized as one of the major drivers of autoimmunity [129] and are the target of highly effective RA therapies such as rituximab [130]. Colocalization of MAA with autoreactive B lymphocytes in lung tissues of

RA-ILD patients [131] further signifies their potential role in the pathogenesis of RA-ILD. While NK cells are considered a bridge between innate and adaptive immune responses [132], targeting cluster 9b could be of interest. Similarly, autoreactive T lymphocytes (perhaps cluster 6b) and cellular phenotypes supporting fibroproliferation with increase in activated fibroblasts with extracellular matrix deposition could be of particular interest in RA-ILD.

In addition to confirming the upregulation of several interferon-induced genes implicated in autoimmunity, complement cascade genes such as *C1ra*, *C1qa*, *C1qb*, *C2*, and *Cfb* representing classical and alternative pathways [133] were identified. *C1s1* was highlighted among the complement cascade genes due to its high expression, and was invariably upregulated in the CIA+ODE group in all 4 cell-sorted neutrophil and monocyte-macrophage populations. *C1* acts as a sensor for self and non-self-recognition and thus plays a major role in self-tolerance [134]. Complement cascade genes are recognized in RA [135] and ILD [136–138], but remain overlooked as therapeutic targets [139]. Holers *et. al.* [135] suggested that RA-ILD has a complement connection, but to our knowledge, this is among the first reports to experimentally identify *C1s1* or complement cascade genes in a RA-inflammatory lung disease model.

There are limitations of this study. In general, animal modeling of arthritis-lung disease interactions are overall limited. The over-expressing TNF-alpha transgenic mice that develop an array of connective tissue diseases has been associated with interstitial lung disease and pulmonary hypertension, particularly in female mice [22]. The arthritic SKG mice develop a cellular and fibrotic interstitial pneumonia [140], but SKG mice do not develop compelling evidence of autoimmunity or arthritis following inhalation injury (i.e tobacco smoke or bleomycin) [140]. Our model system utilizing complex agriculture dust exposure-induced airway inflammation in the setting of CIA would be representative of populations whereby inhalant airborne toxicants are recognized risk factors for RA and RA-lung disease. Cigarette smoke exposure is the most-well-environmental established risk for RA, and occupational inhalant exposures (e.g. work exposures in farming, construction, mining, warehouse environments) have been increasingly associated with risk of disease development, particularly among men [112, 141, 142]. Moreover, it has previously been reported that ODE exposure induced citrullinated and malondialdehyde-acetaldehyde (MAA) modified proteins in the lung tissue [114], antigens that have been implicated in RA pathogenesis. The advantage of this modeling system is that disease development is dependent on airborne biohazard exposures and can be readily modified to investigate other types of exposures. These current studies were focused on male mice because previous work demonstrated that male mice were profoundly more susceptible to CIA+ODE induced adverse effects as compared to female mice [11]. Moreover, it is also recognized that non-arthritic male mice are susceptible to inhalant endotoxin-induced bone loss whereas female mice were protected and that this protection was dependent upon estrogen [143]. Future studies may be warranted to investigate how single airborne biohazard exposures (e.g. endotoxin, pollutants) in the setting of arthritis induction affect lung-arthritis disease as well as specific hormone factors including testosterone, progesterone, and estrogen. Although lung DCs were noted to be limited in this mouse modeling system, future studies could also investigate the role of lung DCs in lung inflammatory disorders associated with RA.

In conclusion, application of scRNA-seq to an animal model combining systemic arthritis induction and environmental inhalant-induced lung inflammation (i.e. RA-associated lung disease) identified unique populations of lung immune cell clusters differentially ascribed to individual treatment conditions. Neutrophil subpopulations and heterogeneous macrophage-monocyte populations were identified in addition to unique genes (interferon-related and complement cascade) that could be contributing to the pathogenesis of RA-associated lung disease. Additionally, this information might inform potential candidates that could be

exploited in future investigations examining targeted interventions and the identification of informative disease biomarkers.

## Supporting information

**S1 Fig. Gating strategy for FACS-based population isolation.** Neutrophils were sorted from lung digests as live, singlets, CD45$^+$, non-lymphocytes, Ly6C$^+$ and Ly6G$^+$. Three monocyte/ macrophage populations were sorted from lung digests as live, singlets, CD45$^+$, non-lymphocytes, Ly6C$^-$ and Ly6G$^-$, and identified as separate populations with variable expression of CD11b and CD11c as: (1) macrophages (CD11c$^{high}$, CD11b$^{variable}$), (2) monocytes-macrophages (CD11c$^{intermediate}$, CD11b$^{high}$), and (3) monocytes (CD11c$^-$, CD11b$^{high}$).
(TIF)

**S2 Fig. Gating strategy for posthoc analysis of myeloid-derived suppressor cells (MDSCs).** Granulocytic (g) MDSCs were identified as live, singlets, CD45$^+$, non-lymphocytes that were Ly6C$^+$ Ly6G$^+$ CD11b$^+$ SSC$^{high}$. Whereas monocytic (m) MDSCs were identified as live, singlets, CD45$^+$, non-lymphocytes that were Ly6G$^-$ CD11b$^+$ Ly6C$^+$ SSC$^{low}$.
(TIF)

**S1 Table. The top 10 genes uniquely identified to neutrophil subtypes with mean UMI count, log2 fold-change and adjusted p value as compared to all other CD45+ lung cell clusters.**
(PDF)

**S2 Table. The top 10 genes uniquely identified to monocyte, macrophage and DC subtypes with average UMI count, log2 fold-change and adjusted p value compared to all other CD45+ lung cell clusters.**
(PDF)

**S3 Table. The top 10 genes uniquely identified to lymphocytes subtypes with average UMI count, log2 fold-change and adjusted p value compared to all other CD45+ lung cell clusters.**
(PDF)

## Acknowledgments

We thank Victoria B. Smith, Samantha D. Wall, Craig L. Semerad in the Flow Cytometry Research Facility at UNMC for technical support for flow cytometry studies. We thank Jennifer L. Bushing in the Sequencing Core at UNMC for assistance with NanoString studies. We also thank Lisa R. Chudomelka for article preparation assistance and submission.

## Author Contributions

**Conceptualization:** Rohit Gaurav, Ted R. Mikuls, Geoffrey M. Thiele, Michael J. Duryee, Bryant R. England, Jill A. Poole.

**Data curation:** Rohit Gaurav, Amy J. Nelson, Meng Niu, James D. Eudy, Austin E. Barry.

**Formal analysis:** Rohit Gaurav, Meng Niu, Chittibabu Guda, James D. Eudy.

**Funding acquisition:** Ted R. Mikuls, Chittibabu Guda, James D. Eudy, Bryant R. England, Jill A. Poole.

**Investigation:** Rohit Gaurav, Jill A. Poole.

**Methodology:** Rohit Gaurav, Amy J. Nelson, Meng Niu, Chittibabu Guda, James D. Eudy.

**Resources:** Chittibabu Guda, Jill A. Poole.

**Software:** Meng Niu, Chittibabu Guda, James D. Eudy.

**Supervision:** Chittibabu Guda, Jill A. Poole.

**Validation:** Rohit Gaurav, Jill A. Poole.

**Visualization:** Rohit Gaurav.

**Writing – original draft:** Rohit Gaurav, Jill A. Poole.

**Writing – review & editing:** Rohit Gaurav, Ted R. Mikuls, Geoffrey M. Thiele, Amy J. Nelson, Meng Niu, Chittibabu Guda, James D. Eudy, Austin E. Barry, Todd A. Wyatt, Debra J. Romberger, Michael J. Duryee, Bryant R. England, Jill A. Poole.

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
