## [Decision Letter · Decision Letter 0]

23 Oct 2020

PONE-D-20-30693

High-Throughput Analysis of Lung Immune Cells in a Murine Model of Rheumatoid Arthritis-Associated Lung Disease

PLOS ONE

Dear Dr. Gaurav,

Thank you for submitting your manuscript to PLOS ONE. After careful consideration, we feel that it has merit but does not fully meet PLOS ONE’s publication criteria as it currently stands. Therefore, we invite you to submit a revised version of the manuscript that addresses the points raised during the review process.

We look forward to receiving your revised manuscript.

Kind regards,

Michal A Olszewski, DVM, PhD

Academic Editor

PLOS ONE

Journal Requirements:

2.At this time, we request that you  please report additional details in your Methods section regarding animal care, as per our editorial guidelines:

(1) Please state the number of mice used in the study

(2) Please describe any steps taken to minimize animal suffering and distress, such as by administering anaesthesia

(3) Please describe the post-exposure care received by the animals, including the frequency of monitoring and the criteria used to assess animal health and well-being.

Thank you for your attention to these requests.

3. Please provide the product number and any lot numbers of the antibodies used for cell sorting in your study.

Additional Editor Comments (if provided):

1) The model presented here appears to be not a true model of RA-associated lung disease, but more a Farmer's Lung Disease complicated with the RA, and thus relevance to actual RA-ALD is questionable.

The title of the manuscript should be corrected to reflect this e.g.: model of swine dust-triggered inflammatory lung disease in mice with RA, to avoid the misrepresentation.

2) Please address comment 3 of Reviewer 1 as the highest priority (clinical score/joint deformation, etc) to determine if S.D.-lung priming affects the course of RA in this model. Negative data woudl be fine

3) Please add some longitudinal data (could be histology as suggested in comments 2 and 6 of Rev. 1)

4) Discuss the limitation of current model and possibly how it compares to other models.

Reviewers' comments:

Reviewer's Responses to Questions

**Comments to the Author**

1. Is the manuscript technically sound, and do the data support the conclusions?

Reviewer #1: Partly

Reviewer #2: Partly

2. Has the statistical analysis been performed appropriately and rigorously? 

Reviewer #1: I Don't Know

Reviewer #2: Yes

3. Have the authors made all data underlying the findings in their manuscript fully available?

Reviewer #1: Yes

Reviewer #2: Yes

4. Is the manuscript presented in an intelligible fashion and written in standard English?

Reviewer #1: Yes

Reviewer #2: Yes

5. Review Comments to the Author

Reviewer #1: Extensive descriptive data about gene expression patterns that define inflammatory cell subsets is presented from a mouse model RA/RA-ILD. It is challenging to understand the implications of this work for human disease for the following reasons;

1- Exposure to pig farm dust is not likely relevant to RA-ILD in >95% of humans with this disease. The SKG mouse is a more compelling model of RA and RA-ILD, that does not require exposure to airway irritants and that has autoantibodies similar to human RA. Why is the model used in this study more appropriate than the SKG mouse? Does it lead to citrullination of lung proteins?

2- The timing of the sample collection is not always clear, but the study appears to be from the late stage of disease. Some longitudinal data might be illuminating,

3- What severity of arthritis was induced? Did the exposure to swine dust affect the course of the arthritis?

4- Only male mice were used, but RA is more common in females (although the ILD can be severe in males with RA), Consider comparing genders.

5- Insufficient information is presented to establish relevance of the data to human RA.

6- Histology of mouse lungs to demonstrate the locations of the cell types identified by RNA profiling would be of interest.

Reviewer #2: In this paper Gaurav et. al. use single-cell RNA sequencing and NanoString analysis to study the lung immune cell populations and transcriptional profiles in a murine model of Rheumatoid arthritis-associated lung disease. The study itself well performed with appropriately statistical analysis. Through descriptive, it provides systemic information and serve as a resource for future studies about the RA associated diseases. However, the reviewer has several major concerns about this study.

1. one major concerns is about the sorting strategy used for isolate immune subsets for Nanostring analysis. In fig 6 in the ODE and CIA+ODE groups, there are cells (which seems to be over-compensated by flow) expressing high level of Ly6G. Are those also neutrophils and should be included in the study? The authors showed more than 30% cells are neutrophils in the control lungs, is this normal?

In fig 7, using CD11c and CD11b markers solely is hard to identify subset like macrophages, mo-macs and monocytes. For example, CD11c is recognized as a DC marker instead of macrophages. These sorting strategies makes the NanoString data hard to interpret.

2. Fig 2, the authors should provide quantitative data about the proportion and absolute number for each subsets. It is hard to find differences virtually and this figure were referred by many places in the manuscript.

6. PLOS authors have the option to publish the peer review history of their article (what does this mean?). If published, this will include your full peer review and any attached files.

Reviewer #1: No

Reviewer #2: No

---

## [Author Response · Author response to Decision Letter 0]

3 Dec 2020

We thank the reviewers and Editor for their thorough review and comments and providing opportunity to improve our manuscript. We have fully addressed comments by the editor and reviewers. Please see the point-by-point response to all the comments in “Response to Reviewers”.

General Comments:

C1. Please state the number of mice used in the study

R1. Thank you. The exact number of mice per experiment has been added to the methods section and the figure legends. The following changes were added to the methods section:

“Arthritis evaluation was assessed on 8 mice per treatment group from 3 independent studies.”

RNA sequencing studies: “For these studies, 2 mice per treatment groups were….”

Flow cytometry sorting/Nanostring studies: “In separate studies with 8 mice per treatment group from 3 independent studies and following mouse euthanasia…”

C2. Please describe any steps taken to minimize animal suffering and distress, such as by administering anaesthesia

R2. Isoflurane was used as anesthesia to minimize suffering and distress while instilling ODE or CIA injections. To the method’s section we added, “All procedures on mice were done under isofluorane to minimize distress.”

C3. Please describe the post-exposure care received by the animals, including the frequency of monitoring and the criteria used to assess animal health and well-being.

Thank you for your attention to these requests.

R3: We added to the method’s section, “After every instillation or injection animals were monitored consistently until they regained consciousness and mobility. They were monitored every day once by the investigators and once by the vivarium staff.” 

C4. Please provide the product number and any lot numbers of the antibodies used for cell sorting in your study.

R4: Thank you for the comment. We have included the catalog and lot numbers for the antibodies used.

Additional Editor Comments:

C1. The model presented here appears to be not a true model of RA-associated lung disease, but more a Farmer's Lung Disease complicated with the RA, and thus relevance to actual RA-ALD is questionable.

The title of the manuscript should be corrected to reflect this e.g.: model of swine dust-triggered inflammatory lung disease in mice with RA, to avoid the misrepresentation.

R1. Thank you for the comment and suggestion. The title has been changed to: “High-Throughput Analysis of Lung Immune Cells in a Combined Murine Model of Agriculture Dust-Triggered Airway Inflammation with Rheumatoid Arthritis.”

Combining C2 and C3. C2)Please address comment 3 of Reviewer 1 as the highest priority (clinical score/joint deformation, etc.) to determine if S.D.-lung priming affects the course of RA in this model. Negative data would be fine.C3. Please add some longitudinal data (could be histology as suggested in comments 2 and 6 of Rev. 1)

Combined responses of R2 and R3. We appreciate the opportunity to present the arthritis inflammatory scores (new Fig 1A) as these were recorded longitudinally (i.e. weekly) in mice utilized for the cell sorting/nanostring studies and were consistent with the previous publication describing the model (Poole, et al. J Bone Min Research 2019). Moreover, we took this opportunity to improve the description of past work in detailing the arthritis/bone deterioration findings (as well as lung histology consequences). Because we utilized all lung tissue for the high-throughput analysis, we obtained lung sections from our previous study that utilized exact same procedure (Poole, et al. J Bone Min Research 2019). We stained lung sections for H&E from each treatment group showing increased inflammation in CIA+ODE (Fig 1B). Sections were also stained with markers to identify macrophages (CD68), neutrophils (MPO), T cells (CD3), and B cells (CD45R) (Figs 1C-D) in the parenchyma, peri-bronchiolar and peri-vasculature regions. Interestingly, in these new studies, we investigated whether inflammatory monocytes/macrophages noted as CCR2+ were recruited to the lung to correspond to the cell sorting/Nanostring analysis studies. Indeed, we now demonstrate this cell, and more importantly, it is increased with CIA and ODE and further potentiated with combined exposures (CIA+ODE) (Fig 1E-F). Collectively, we added a new Figure 1A-E and rearranged the number of the subsequent figures for improved flow of the manuscript. These major revisions to the results and methods and discussion are specifically outlined below: 

Methods section: “Arthritis evaluation: Arthritis inflammatory scores were assessed weekly using the semiquantitative, mouse arthritis scoring system provided by Chondrex (www.chondrex.com) as previously described (13). Scores range from 0 (no inflammation) to 4 (erythema and severe swelling encompassing ankle, foot, and digits).”

Methods section: “Lung Histopathology: Lung sections of Sham, CIA, ODE, and CIA+ODE treatment groups previously obtained (13) were stained with H&E or with anti-CD3 (1:100, Cat#ab5690, Lot#GR3356033-2), anti-CD68 (1:50, Cat#ab31630, Lot#GR3305929-3), and anti-MPO (1:25, Cat#ab9535, Lot#GR331736-4) from Abcam (Cambridge, MA, USA), anti-CD45R (1:40, Cat#14-0452-82, Lot#2178350) from Invitrogen (Grand Island, NY, USA), and anti-CCR2 (1:100, Cat# NBP267700, Lot# HMO537) from Novus Biologicals (Centennial, CO, USA). Cross absorbed (H+L) goat anti-rabbit (Cat#A32731, Lot#UK290266), goat anti-mouse (Cat#A32727, Lot#UL287768) and goat-anti rat (Cat#A21434, Lot#2184321) from Thermo Fisher, Grand Island, NY, USA) were used at 1:100 dilution as secondary antibodies. Slides were mounted with VECTASHIELD® Antifade Mounting Medium with DAPI (Cat#H-1200, Lot#ZG1014, Burlingame, CA, USA) and visualized under Zeiss fluorescent microscope.

Results section: “Agriculture (swine) exposure-related ODE-induced airway inflammation coupled with arthritis modeling. Consistent with the previously published study describing this model system (13), the highest arthritis inflammatory scores were demonstrated for CIA+ODE at 5 weeks (Fig 1A). Throughout the 5 weeks, arthritis inflammatory scores were recorded weekly, and as early as 1 week, there were increases in arthritis inflammatory scores in CIA+ODE and CIA alone as compared with Sham. At 4 weeks, there were significant increases in ODE alone. Previous lung pathology investigations (13) reported increases in alveolar and bronchiolar compartment inflammation and increases in cellular aggregates with ODE and CIA+ODE as compared to sham, but the number and size of cellular aggregates were reduced in CIA+ODE as compared to ODE alone. Instead, pro-lung fibrosis mediators including lung hyaluronan and fibronectin were increased in CIA+ODE as compared to ODE alone (13). In these current studies, lung sections from the previous study were stained with H&E from each treatment group showing increased inflammation in CIA+ODE (Fig 1B). Sections were also stained with markers to identify macrophages (CD68), neutrophils (MPO), T cells (CD3), and B cells (CD45R) (Figs 1C-D) distributed in the parenchyma, peribronchiolar and perivascular regions. Infiltration of recruited CCR2+ inflammatory monocytes were also demonstrated in the CIA and the ODE groups, and these cells were further potentiated in the CIA+ODE group (Figs 1E-F). These studies lay the foundation for current studies delineating cellular and gene determinations through high-throughput analysis.”

Discussion section: “Correspondingly, infiltrative CCR2+ inflammatory monocytes were demonstrated with CIA and ODE with highest expression in combined exposures. Future studies could determine whether targeting this cell population results in a reduction of disease manifestations.” 

C4. Discuss the limitation of current model and possibly how it compares to other models.

R4. Thank you for the opportunity to compare and contrast our current model system with other autoimmune and/or arthritis combined with lung disease models. We added this following paragraph to the discussion section:

“There are limitations of this study. In general, animal modeling of arthritis-lung disease interactions is overall limited. The over-expressing TNF-alpha transgenic mice that develop an array of connective tissue diseases has been associated with interstitial lung disease and pulmonary hypertension, particularly in female mice (19). The arthritic SKG mice develop a cellular and fibrotic interstitial pneumonia (133), but SKG mice do not develop compelling evidence of autoimmunity or arthritis following inhalation injury (i.e tobacco smoke or bleomycin) (133). Our model system utilizing complex agriculture dust exposure-induced airway inflammation in the setting of CIA would be representative of populations whereby inhalant airborne toxicants are recognized risk factors for RA and RA-lung disease. Cigarette smoke exposure is the most-well-established environmental risk for RA, and occupational inhalant exposures (e.g. work exposures in farming, construction, mining, warehouse environments) have been increasingly associated with risk of disease development, particularly among men (111, 140, 141). Moreover, it has previously been reported that ODE exposure induced citrullinated and malondialdehyde-acetaldehyde (MAA) modified proteins in the lung tissue (107), antigens that have been implicated in RA pathogenesis. The advantage of this modeling system is that disease development is dependent on airborne biohazard exposures and can be readily modified to investigate other types of exposures. These current studies were focused on male mice because previous work demonstrated that male mice were profoundly more susceptible to CIA+ODE induced adverse effects as compared to female mice (13). Moreover, it is also recognized that non-arthritic male mice are susceptible to inhalant endotoxin-induced bone loss whereas female mice were protected and that this protection was dependent upon estrogen (134). Future studies may be warranted to investigate how single airborne biohazard exposures (e.g. endotoxin, specific pollutants) in the setting of arthritis induction affect lung-arthritis disease as well as specific hormone factors including testosterone, progesterone, and estrogen.” 

Reviewer Comments:

Reviewer 1.

C1. Exposure to pig farm dust is not likely relevant to RA-ILD in >95% of humans with this disease. The SKG mouse is a more compelling model of RA and RA-ILD, that does not require exposure to airway irritants and that has autoantibodies similar to human RA. Why is the model used in this study more appropriate than the SKG mouse? Does it lead to citrullination of lung proteins?

R1. The reviewer raises important questions about our animal modeling system, particularly in comparison to the SKG mouse model (please see R4 in response to the Editor’s comments above). We agree that the SKG mice develop a cellular and fibrotic interstitial pneumonia, but in contrast, the SKG mice do not develop compelling evidence of autoimmunity or arthritis following inhalation injury (i.e. tobacco smoke or bleomycin). The advantage of our model is that it does require exposure to airway irritants, which we highlight is directly relevant to human disease as airborne toxicants are established risk factors for RA and RA-ILD. Cigarette smoke is the most well-established risk factor, and occupational exposures (e.g. work exposures in farming, construction, mining, warehouse environments) have been increasingly associated with risk of disease development, particularly among men. Thus, a “second hit” by an airborne biohazard likely represents a critical factor in susceptibility to human disease manifestations. It would be interesting in future studies to investigate single agents such as endotoxin, peptidoglycan, particulates, diesel exhaust, among others by leveraging this model system. We have previously published that anti-cyclic citrullinated peptide IgG antibodies are increased with CIA+ODE, which does not occur with CIA or ODE alone. We have also reported that ODE exposure induced citrullinated and malondialdehyde-acetaldehyde (MAA) modified proteins in the lung tissue. To improve the manuscript, we have added a discussion paragraph of lung-arthritis mouse modeling and provided information from our past studies to the introduction. The following was added:

Introduction: “…the combination of the collagen-induced arthritis (CIA) model with a model of airborne biohazard exposure (e.g. agriculture related-organic dust extract/ODE) resulted in augmented arthritis inflammatory score and bone deterioration, increased systemic autoimmunity with increased anti-cyclic citrullinated peptide IgG antibodies, and promotion of pre-fibrotic inflammatory lung changes in mice (11) consistent with RA-associated lung disease pathophysiology.”

Discussion: See new limitation paragraph for discussion under Response 4 (R4) to Comment 4 (C4) by Editor. 

C2. The timing of the sample collection is not always clear, but the study appears to be from the late stage of disease. Some longitudinal data might be illuminating.

R2. First, we would like to clarify that the timing of the sample collection was consistent across experiments. Specifically, the murine treatment groups of Sham (saline injection/saline inhalation), CIA (CIA injection/saline inhalation), ODE (saline injection/ODE inhalation), and CIA+ODE (CIA injection/ODE inhalation) were done in parallel with euthanization occurring 5 weeks after initiation of treatments. This clarification was added to the methods section. Specifically, we added: “Mouse treatment groups were ran in parallel with euthanization occurring 5 weeks after initiation of treatments.” 

C3. What severity of arthritis was induced? Did the exposure to swine dust affect the course of the arthritis?

R3. We thank the reviewer for the opportunity to provide information regarding the arthritis inflammatory scores that were assessed weekly over the 5-week treatment course in the animals subjected to the nanostring analysis studies. We provided this information in a new Figure 1A. Moreover, we are also taking this opportunity to provide further background on the prior publication of this co-exposure model that highlighted the arthritis/bone deterioration consequences in the introduction (see response to comment 1). 

Results section: “Consistent with previously published study describing this model system (13), the highest arthritis inflammatory scores were demonstrated for CIA + ODE at 5 weeks (Figure 1A). Throughout the 5 weeks, arthritis inflammatory scores were recorded weekly, and as early as 1 week, there were increases in arthritis inflammatory scores in CIA+ODE and CIA alone as compared with sham. At 4 weeks, there were significant increases in ODE alone.”

C4. Only male mice were used, but RA is more common in females (although the ILD can be severe in males with RA), Consider comparing genders.

R4. The reviewer pointed out an accurate observation, and in our previous publication, we compared and reported on the differences between the male and female mice. There was a clear sex difference. The male mice were profoundly more susceptible to adverse effects in this co-exposure (CIA+ODE) than the female mice. Importantly, the sex-based differences observed in this model appear mirror those observed in RA with epidemiologic data suggesting that men are more susceptible to RA-related interstitial lung diseases than women (Kelly CA, et al. Rheumatology 2014). In addition, we had also previously found that non-arthritic male mice were susceptible to inhalant endotoxin-induced bone loss as compared to female mice. Moreover, oophorectomized female mice gained bone loss susceptibility to inhalant endotoxin challenges and that estrogen replacement reversed the endotoxin-mediated bone loss in both female and male. In this current study we focused on the male mice because they have the profound phenotype to investigate high-throughput studies. As this is a very important point, we added this information to the discussion (see paragraph under response to editor comments; R4).

C5. Insufficient information is presented to establish relevance of the data to human RA.

R5. With the additional information and background information specific to the previous publication describing this airborne biohazard exposure model, we feel the justification is now present to support the current study. Please see comprehensive response 1 to comment 1.

C6. Histology of mouse lungs to demonstrate the locations of the cell types identified by RNA profiling would be of interest.

R6. We agree with the reviewer that providing lung histology images will improve the current study focusing on high-throughput analysis. As the whole lungs were processed to optimize lung cell recovery, we sectioned lungs from the previous published study describing the combined exposure model. Using these lung sections, we present new H&E images and stained sections for CD68 (macrophages), CD3 (T-cells), CD45R (B220; B-cells) and MPO (neutrophils), and CCR2+ inflammatory monocytes (new Figure 1B-E).

Reviewer 2.

C1. one major concerns is about the sorting strategy used for isolate immune subsets for Nanostring analysis. In fig 6 in the ODE and CIA+ODE groups, there are cells (which seems to be over-compensated by flow) expressing high level of Ly6G. Are those also neutrophils and should be included in the study? The authors showed more than 30% cells are neutrophils in the control lungs, is this normal?In fig 7, using CD11c and CD11b markers solely is hard to identify subset like macrophages, mo-macs and monocytes. For example, CD11c is recognized as a DC marker instead of macrophages. These sorting strategies makes the NanoString data hard to interpret.

R1. Thank you for the excellent comments and the opportunity to clarify methods and discuss future opportunities. In Fig 6 (now Fig 7), we sorted cells based on Ly6Clow or high and Ly6G+ to capture the neutrophil population consistent with all 4 groups. Due to technical limitation of ability to sort for maximum of 4 cell populations at one time, we did not capture Ly6C- cells as that population was not sufficiently represented in Sham or CIA for comparison studies. We recognize that future studies could be interesting focusing on that population that appears to be driven by ODE, and we have added this suggestion to the discussion as limitation.

Discussion section: “In the cell sorting/NanoString studies, neutrophils were identified and limited to Ly6C+ Ly6G+ cells based upon equipment limitations of simultaneous 4-cell capture ability. Future studies to include the Ly6G+ Ly6C- cells that were predominately observed with ODE and CIA+ODE could also be informative.”

Next, we clarify that the percentage of neutrophils are represented as a subset and not of total cells, which is determined in total CD45+ cells. This percentage is in DBA/1J mice but is consistent with previous literature, reflecting a population of resident neutrophils in lungs (PMID: 26881904). We have added the following text in results:

Results section: “Lung neutrophils were isolated by fluorescence activated cell sorting (FACS) based on traditional cell surface markers as percent of CD45+ cells that were Ly6C+ Ly6Ghigh (Figs 7A and S1). Sham represented resident neutrophils in the lungs at baseline (101).”

We also agree with the reviewer that non-lung DCs predominately express CD11c, particularly myeloid DCs. In the lung, macrophages highly express CD11c (PMID: 32544188, PMID: 16362884). The lung DC percentage as identified with Siglech expression was extremely low. Additionally, the expression level of CD11c in lung DCs identified by RNAseq was very low compared to the lung macrophages. Importantly, this gating strategy is consistent with prior reports by us and others for identification of lung macrophage-monocytes(PMID: 282 57233, PMID: 27190063, PMID: 24325577, PMID: 31693392). We added the following to the method’s section: “This gating strategy for monocyte/macrophage populations is consistent with previous reports by us and others (18-21).” 

Finally, because lung DC’s are also an important cell population, we added to the discussion the following: “Although lung DCs were noted to be limited in this mouse modeling system, future studies could also investigate the role of lung DCs in lung inflammatory disorders associated with RA.”

C2. Fig 2, the authors should provide quantitative data about the proportion and absolute number for each subsets. It is hard to find differences virtually and this figure were referred by many places in the manuscript.

R2. Thank you for this excellent suggestion to further improve the quality and understanding of our data. We have now calculated exact numbers from each subset and added this data to the figure (new Fig 3C) showing cell numbers in each sub-cluster among treatment groups. Please note that the old Fig 2A-B is now Fig 3A-C with appropriate modifications to the figure legend. We have also added the following text in the results section: “Based on the shifts observed in the cell populations among treatment groups, a manual sub-clustering was performed to delineate exact number of cells distributed among the sub-clusters (Fig 3C).”

---

## [Decision Letter · Decision Letter 1]

21 Dec 2020

High-Throughput Analysis of Lung Immune Cells in a Combined Murine Model of Agriculture Dust-Triggered Airway Inflammation with Rheumatoid Arthritis

PONE-D-20-30693R1

Dear Dr. Gaurav,

We’re pleased to inform you that your manuscript has been judged scientifically suitable for publication and will be formally accepted for publication once it meets all outstanding technical requirements.

Kind regards,

Michal A Olszewski, DVM, PhD

Academic Editor

PLOS ONE

Additional Editor Comments (optional):

Reviewers' comments:

Reviewer's Responses to Questions

**Comments to the Author**

1. If the authors have adequately addressed your comments raised in a previous round of review and you feel that this manuscript is now acceptable for publication, you may indicate that here to bypass the “Comments to the Author” section, enter your conflict of interest statement in the “Confidential to Editor” section, and submit your "Accept" recommendation.

Reviewer #1: All comments have been addressed

Reviewer #2: All comments have been addressed

2. Is the manuscript technically sound, and do the data support the conclusions?

Reviewer #1: Yes

Reviewer #2: (No Response)

3. Has the statistical analysis been performed appropriately and rigorously? 

Reviewer #1: Yes

Reviewer #2: (No Response)

4. Have the authors made all data underlying the findings in their manuscript fully available?

Reviewer #1: Yes

Reviewer #2: (No Response)

5. Is the manuscript presented in an intelligible fashion and written in standard English?

Reviewer #1: Yes

Reviewer #2: (No Response)

6. Review Comments to the Author

Reviewer #1: Thank you for addressing the issues that I raised.

This is an interesting paper that should stimulate further work in this area.

Reviewer #2: (No Response)

7. PLOS authors have the option to publish the peer review history of their article (what does this mean?). If published, this will include your full peer review and any attached files.

Reviewer #1: No

Reviewer #2: No

---

## [Editor Report · Acceptance letter]

3 Feb 2021

PONE-D-20-30693R1 

High-Throughput Analysis of Lung Immune Cells in a Combined Murine Model of Agriculture Dust-Triggered Airway Inflammation with Rheumatoid Arthritis 

Dear Dr. Gaurav:

I'm pleased to inform you that your manuscript has been deemed suitable for publication in PLOS ONE. Congratulations! Your manuscript is now with our production department. 

Kind regards, 

on behalf of

Dr. Michal A Olszewski 

Academic Editor

PLOS ONE